# A Longitudinal Analysis of the Internal Rotation and Shift (IRO/Shift) Test Following Arthroscopic Repair of Superior Rotator Cuff Lesions

**DOI:** 10.3390/jpm12122018

**Published:** 2022-12-07

**Authors:** René Schwesig, George Fieseler, Jakob Cornelius, Julia Sendler, Stephan Schulze, Souhail Hermassi, Karl-Stefan Delank, Kevin Laudner

**Affiliations:** 1Department of Orthopedic and Trauma Surgery, Martin-Luther-University Halle-Wittenberg, 06120 Halle, Germany; 2Clinic for Orthopedic and Trauma Surgery, Sports Medicine, Clinic Hannoversch Münden, 34346 Hannoversch Münden, Germany; 3Physical Education Department, College of Education, Qatar University, Doha 2713, Qatar; 4Department of Health Sciences, University of Colorado Colorado Springs, Colorado Springs, CO 80918, USA

**Keywords:** arthroscopic surgery, clinical test, diagnostic accuracy, shoulder, supraspinatus

## Abstract

Although the use of clinical tests to diagnose superior rotator cuff pathology is common, there is paucity in the research regarding the accuracy of such tests following arthroscopic repair. The aim of this study was to determine the accuracy of the IRO/Shift test compared to the Jobe test at 3 months and 6 months post-surgery for superior rotator cuff repair. Arthroscopic repair was conducted on 51 patients who were subsequently seen for clinical evaluation at 3 and 6 months following surgery. At 3 months post-surgery only 27% of the patients had a negative IRO/Shift test and 18% had a negative Jobe test. However, at 6 months 88% of the patients presented with a negative IRO/Shift test and 61% a negative Jobe test. When compared to each other, the IRO/Shift test and the Jobe test had 90% agreement pre-operatively, 71% agreement at 3 months post-surgery, and 67% agreement at 6 months. These results demonstrate that the accuracy of the IRO/Shift test and the Jobe test improved between 3 and 6 months following arthroscopic surgery of the superior rotator cuff, with the IRO/Shift test having better accuracy.

## 1. Introduction

The accurate diagnosis of superior rotator cuff lesions is a challenging process. Clinicians often use a combination of techniques including various special tests such as the Jobe test, shoulder shrug sign, full can test, shoulder drop arm test, and others, as well as advanced imaging to maximize their clinical impression [1]. Unfortunately, there is still some degree of error that comes with these clinical tests, with some tests reporting strong sensitivity, but poor specificity or vice versa [2,3,4,5]. Previous research has also investigated the use of predictive models, which combine various special tests for increased accuracy [3,4]. This has led to many reports stressing the importance of using multiple tests to increase diagnostic accuracy, rather than relying on one test [2,6,7,8]. As such, clinicians are continuously trying to stay innovative in what procedures they use during their clinical examinations, which will allow them to better isolate the desired soft tissue structure, or structures, such as the supraspinatus tendon and therefore, increase the accuracy of their diagnoses. Furthermore, the recurrent application of clinical tests following surgery and rehabilitation is useful to supervise the process of convalescence. The conversion of a pathological clinical test to negative results over time demonstrates the appropriate recovery of a torn and surgically repaired tendon. 

One of the more recent tests developed for the assessment of superior rotator cuff pathology, the internal rotation and shift test (IRO/Shift test), was originally described by Fieseler et al. [9]. This test has been shown to be both a reliable and valid technique for identifying superior rotator cuff lesions [9,10,11]. More specifically, these authors reported an intrarater reliability of ICC = 0.73 and an interrater reliability of ICC = 0.89, as well as 92% sensitivity, 67% specificity, 86% positive predictive value, and 80% negative predictive value. Subsequent research also found that the IRO/Shift test was comparable to the Jobe test [12], which has been viewed as the gold standard of testing for these types of lesions [2,5]. This study reported the IRO/Shift test to have 96% sensitivity, 50% specificity, 73% positive predictive value, 91% negative predictive value, and an accuracy of 77%, while the Jobe test had 89% sensitivity, 60% specificity, 76% positive predictive value, 80% negative predictive value, and an accuracy of 77%. Although this previous work has demonstrated that the IRO/Shift test is an accurate and valid tool in the diagnosis of this pathology, there is paucity in the research regarding the outcomes of using this technique among post-operative patients. Understanding the usefulness of this special test throughout the post-surgical rehabilitation process could be a critical tool for physicians and physiotherapists as they attempt to accurately assess the healing of the tendon and its correlation with the physical demands of the rehabilitation process. Therefore, the purpose of the current study was to determine how long following the arthroscopic repair of a superior rotator cuff lesion would patients present with a positive IRO/Shift test. More specifically, this study examined the accuracy of using the IRO/Shift test compared to the Jobe test at 3 months and 6 months post-surgery for superior rotator cuff repair.

## 2. Materials and Methods

### 2.1. Subjects

Eighty-seven subjects (51 male, 36 female) volunteered to participate in this study (age: 53.6 ± 12.5 years; height: 1.76 ± 0.09 m; weight: 83.0 ± 15.3 kg, BMI: 26.7 ± 3.51 kg/m^2^). All subjects were 18 years of age or older (age range: 20.9–74.7 years) and presented with a radiologically confirmed structural lesion of the superior rotator cuff, pain, and persistent malfunction of the glenohumeral joint. Prior to their introduction to this study, all subjects had completed a conservative rehabilitation program prescribed by their general physician, conducted under the supervision of a physiotherapist, which resulted in no reduction in their shoulder symptoms. During or after this conservative treatment, magnetic resonance imaging (MRI) was then determined to be necessary to detect the extent of structural pathology and the reasoning for their lack of improvement. Upon review of the MRIs, by an experienced radiologist, structural lesions of the supraspinatus were identified. Following the MRI, all patients were then referred to a single shoulder unit for specialized examination and to determine if surgical intervention was indicated.

Exclusion criteria included a restricted shoulder range of motion, non-traumatic glenohumeral instability, fracture/osseous pathology, acute dislocation, glenohumeral arthrosis, or arthritis. Prior to any data collection, all subjects provided consent as approved by the ethical committee of Martin-Luther-University Halle-Wittenberg (approval number: 2018-05). In total, 59% (51/87) of the patients presented with a radiologically diagnosed superior rotator cuff lesion pre-operatively, which was confirmed intra-operatively. Arthroscopic repair was then conducted on these 51 patients. All surgeries were exclusively performed by an experienced orthopedic shoulder surgeon with more than 20 years of shoulder surgery experience and who performs more than 200 arthroscopic or open surgical procedures per year.

### 2.2. Procedures

This research study used a prospective, blinded study design (Figure 1).

During the initial (pre-operative) clinical examination, two examiners (a surgeon and a resident) performed both the IRO/Shift test and the Jobe test to determine the integrity of the superior rotator cuff tendon. Following arthroscopic repair, each subject returned to the resident who performed the same clinical examinations in the 3-month and 6-month post-operative time periods. During the 3- and 6-month examinations, the resident was blinded to the patient’s status, meaning the resident was unaware of their surgical status.

The IRO/Shift test was performed in accordance with Fieseler et al. [9]. For this test, patients stood in a relaxed position and actively moved their involved shoulder into internal rotation and adduction, by rotating the arm behind their back and then sliding their hand superiorly along their spine, attempting to reach the highest or most superior vertebral spinous process. At their end range of active motion, the clinician then provided a subsequent passive movement of their shoulder into further internal rotation and adduction (i.e., moving the arm and hand more superiorly up the spine) until the end range of passive motion. If increased pain and avoidance was present during this end range of passive motion, the test was considered positive. However, the clinician then needed to rule out possible involvement of the long head of the biceps tendon. If the long head of the biceps tendon was found to be involved (e.g., a positive O’Brien test), then the IRO/Shift test was considered negative for superior rotator cuff pathology. If the long head of the biceps tendon was not found to be involved, then the IRO/Shift test was considered positive.

### 2.3. Statistical Analysis

All statistical analyses were conducted using SPSS version 28.0 for Windows (IBM, Armonk, NY, USA). Arthroscopic findings were used as the gold standard to calculate the sensitivity and specificity (95% confidence interval) of both the IRO/Shift test and the Jobe test. The percentage of positive tests at pre-operation, and 3 and 6 months post-surgery were calculated and reported for both the IRO/Shift test and the Jobe test. The accuracy of the IRO/Shift test was compared to the Jobe test using a Chi-squared analysis. The observed accuracy (percentage) of both tests was defined as consistent negative and positive results and assessed using a four-field table. In this context, a Chi-squared analysis was used to determine the relationship between positive and negative findings for each test in order to illustrate the recovery process.

## 3. Results

The sensitivity of the IRO/Shift test (92%) and the Jobe test (94%) among the 87 patients who initially received a clinical examination prior to any surgical intervention was comparable with each other (Table 1). With respect to specificity, prior to shoulder surgery, a difference of 10% (IRO/Shift test: 68% vs. Jobe test: 78%) was calculated when confirmed with MRI findings.

Among the 51 patients who underwent rotator cuff repair, 22 had a single row repair, 16 had a double row repair, and 13 had debridement only. Figure 2 displays the number of positive IRO/Shift tests longitudinally based on surgical procedure.

Among the 36 patients who underwent arthroscopic surgery, but did not have a rotator cuff tear, 15 had a positive IRO/Shift test and 8 had a positive Jobe test pre-surgery. At 3 months post-surgery there were six positive IRO/Shift and six positive Jobe tests, while at 6 months there was only one positive IRO/Shift and two positive Jobe tests.

In total, 47 of the 51 surgical patients presented with a positive IRO/Shift test pre-surgery, while 48 presented with a positive Jobe test (Figure 3). At 3 months post-surgery there were 31 positive IRO/Shift tests and 36 positive Jobe tests. Meaning only 39% (20/51) had a negative IRO/Shift test and 29% (15/51) had a negative Jobe test at 3 months post-surgery. However, at 6 months post-surgery there were only 5 remaining positive IRO/Shift tests and 18 positive Jobe tests. This resulted in 90% of the patients (46/51) presenting with a negative IRO/Shift test and 65% (33/51) of the patients presenting with a negative Jobe test at 6 months post-surgery. When compared to each other over time, the observed accuracy between the tests decreased. The IRO/Shift test and Jobe test had 90% agreement (i.e., accuracy) pre-operatively, 71% agreement at 3 months post-surgery, and 67% agreement at 6 months (Table 2).

The only statistically significant finding was at 3 months post-surgery (Chi-Squared: 6.72, *p* = 0.010). In relation to the healing process, the number of negative matched findings increased sharply from *n* = 1 (pre-surgery) to *n* = 10 (3-months post-surgery) and *n* = 31 (6 months post-surgery). Conversely, the number of concurrent positive findings decreased from *n* = 45 to *n* = 26 and *n* = 3 during this observation period (Table 2).

## 4. Discussion

The IRO/Shift test has been shown to be a reliable and valid technique for assessing the presence of superior rotator cuff pathology. However, no research has investigated the use of this special test following arthroscopic rotator cuff repair alongside using a longitudinal approach to guide the rehabilitation process and tendon recovery. The results of this study are the first to compare the use of the IRO/Shift test and the Jobe test at 3 months and 6 months post-surgery. These results demonstrate that patients with superior rotator cuff repair present with progressively negative IRO/Shift test and Jobe test findings between 3 and 6 months post-surgery.

Following arthroscopic repair, rotator cuff tendons need a substantial amount of time to heal, with most retears occurring between 6 and 26 weeks (primarily between 12 and 26 weeks) following surgical repair [13]. At 3 months post-surgery, 61% of the patients in the current study still had a positive IRO/Shift test, while 71% still had a positive Jobe test. By 6 months, those positive tests fell to only 10% for the IRO/Shift test and 35% for the Jobe test. This may be explained by the timing of tendon repair and regeneration during the healing process, as well as different forces exerted between the two clinical tests. The Jobe test emphasizes a lever load testing strength via an extended arm placing traction on the tendon, while the IRO/Shift test places a tensile load on the tendon without leverage. Voleti et al. [14] noted that tendon stiffness and collagen reorganization continue to increase from 2 to 16 weeks post-surgery in rats with supraspinatus repair. These authors also reported that less than 10% of the supraspinatus recapitalizes compared to the non-injured limb at 12 weeks post-surgery. Some research has shown that this remodeling phase, where the collagen aligns with stress/strain, predominantly occurs from 1 to 2 months post injury/surgery and can continue for more than 12 months [15]. This may also help explain why the IRO/Shift test, which applies tensile stress to the tendon, had slightly better accuracy following surgery than the Jobe test, which assesses the contractile strength of the muscle.

The differences in stress placed on the rotator cuff tendons by the IRO/Shift and the Jobe tests may potentially explain the divergent correlations of the two tests’ accuracy related to the time of examination throughout the pathological process (pre- and post-operatively) (Table 2). From a statistical point of view, the observed accuracy between both tests steadily decreased from 90% (pre-operatively) to 71% (3 months post-operatively) and 67% (6 months post-operatively). The only statistically significant relationship was found during the second examination (3 months post-surgery) (Chi-squared: 6.72; *p* = 0.010). From a biological and orthopedic perspective, this relationship in the 3-month post-surgery period may have occurred due to the biological demand placed on the tendon while it was still healing. Conversely, during the 6-month examination, the difference in tissue loads between tests may have been more prominent because of the surgically fixed tendon to the footprint and the ability to resist tensile force produced by the IRO/Shift test, and the lack of contractile muscle strength generated by an active movement of the arm during the Jobe test. These findings also support Young’s modulus of elasticity, which notes that following injury collagen is largely disorganized and has a lower resistance to tension and compression. However, throughout the remodeling phase, elasticity continually increases to roughly 80% of the contralateral tendon, while the cross-sectional area and viscoelastic phase angle (tensile strength) are 3 and 1.5 times greater, respectively [16].

In accordance with tendon healing, previous research has shown that multiple functional characteristics significantly improve between 3 and 6 months following arthroscopic rotator cuff repair. For example, He et al. [17] investigated the clinical outcomes of 89 patients who underwent arthroscopic rotator cuff repair using the Southern California Orthopedic Institute row method and found that visual analog scale scores, University of California Los Angeles (UCLA) scores, and Constant–Murley scores, as well as abduction and forward flexion range of motion were all better at 6 months compared to 3 months. Similarly, other research has demonstrated that most improvements in shoulder range of motion (flexion and external rotation), pain, and function occurred by 6 months post-surgery [18,19]. The results of the current study support these previous findings and provide further insight into the use of clinical testing following arthroscopic repair.

In contrast to the IRO/Shift test, which stresses tension on the superior rotator cuff, the Jobe test emphasizes muscle contractile strength to keep the shoulder in a position of internal rotation and elevation. As such, the authors of this study hypothesized that the most of the patients would present with a negative Jobe test 6 months following surgery. This was based on previous studies which have shown increases in muscular strength around the 6-month post-surgery period [19,20]. Unfortunately, only 65% (31/51) of the patients presented with a negative Jobe test at 6 months post-surgery, compared to 90% of the patients with a negative IRO/Shift test. This suggests that test conversion of the IRO/Shift may allow for a quicker return to negative findings following arthroscopic repair due to the stretch and tensile load placed on the tendon compared to the Jobe test. Regardless, clinicians should be discouraged from relying on only one test for diagnosis [2,6,7,8]; therefore, the authors of this study recommend the use of both tests for pre-operative diagnosis and as a tool for post-operative follow-up during rehabilitation and tendon healing.

There are a few limitations of this study worth noting. First, the research participants were relatively young (age: 54 ± 13 years, 33% of patients were younger than 50 years). Previous research has shown that post-operative outcomes can decrease as age increases [21]. Therefore, care should be taken when interpreting these results among older patients. Next, this study identified that negative findings improved for both the IRO/Shift and the Jobe test between 3 and 6 months post-surgery but did not specify an exact time period over this 3-month period. Similarly, the results of this study looked at the short-term outcomes using these clinical tests. Future research should investigate the accuracy of these tests during longer intervals (e.g., 9 months, 12 months, and 24 months post-surgery). Similarly, future research should investigate if patients who still present with a positive IRO/Shift test at the end of the follow-up period (24 months) demonstrate a recurrence of tears or lack of reparative fusion on reimaging. Tendon healing was not monitored post-operatively, so it is impossible to conclude if all tendons were healing at a similar pace. However, all patients that presented with a negative IRO/Shift or Jobe test reported pain-free shoulders with good range of motion and function during daily activities.

## 5. Conclusions

The observed accuracy of the IRO/Shift test and the Jobe test decreased during the observational period (pre-surgery, 3 months post-arthroscopic surgery, and 6 months post-arthroscopic surgery of the superior rotator cuff), depending on the different requirements of the tests and the healing process. This may be due to better alignment of the tensile load accumulated during the IRO/Shift test throughout the healing process. Regardless, both tests demonstrated a high level of sensitivity and specificity. Therefore, these tests should be considered in the post-operative evaluation of superior rotator cuff pathology and during the rehabilitation process.

## Figures and Tables

**Figure 1 jpm-12-02018-f001:**
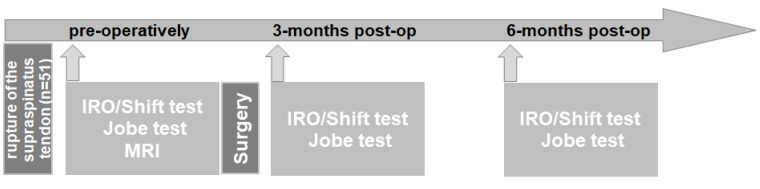
Prospective, blinded study design.

**Figure 2 jpm-12-02018-f002:**
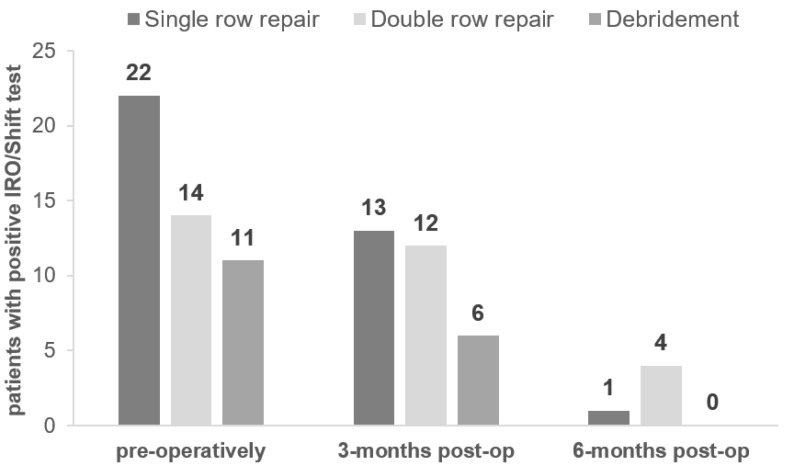
Number of positive IRO/Shift tests longitudinally based on surgical procedure (*n* = 51).

**Figure 3 jpm-12-02018-f003:**
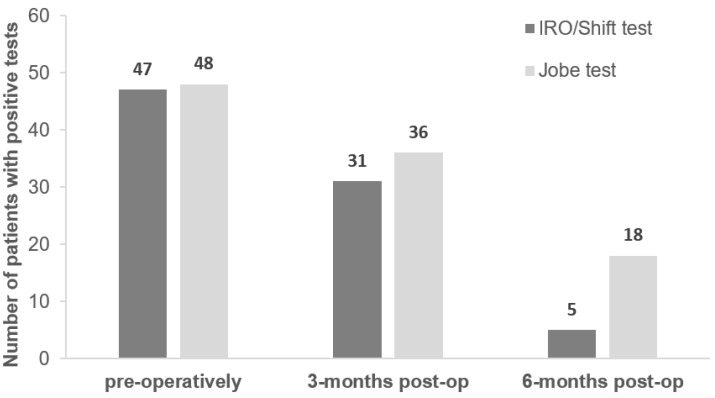
Number of positive IRO/Shift tests and Jobe tests longitudinally. Patients (*n* = 51) with arthroscopically confirmed superior rotator cuff lesions.

**Table 1 jpm-12-02018-t001:** IRO/Shift and Jobe test performance prior to shoulder surgery (confirmed with MRI).

Test	Sensitivity (95% CI)	Specificity (95% CI)
IRO/Shift (%)	92 (87–100)	68 (51–85)
Jobe (%)	94 (88–100)	78 (64–91)

IRO/Shift: internal rotation and shift; CI: confidence interval.

**Table 2 jpm-12-02018-t002:** Observed accuracy (Jobe vs. IRO/Shift) depending on examination points for the injured sample (*n* = 51).

Time Point	Observed Accuracy			Chi-Squared Test (*p*-Value)
	Negative (*n*)	Positive (*n*)	%	
Pre-operative	1	45	90	2.87 (0.091)
3 months post-op	10	26	71	**6.72 (0.010)**
6 months post-op	31	3	67	1.48 (0.224)

IRO/Shift: internal rotation and shift; post-op: post-operative. Significant relationship is bolded.

## Data Availability

Please contact the corresponding author for inquiries regarding study data.

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
