# Peer review of "A Longitudinal Analysis of the Internal Rotation and Shift (IRO/Shift) Test Following Arthroscopic Repair of Superior Rotator Cuff Lesions"

_jpm, 2022, doi:10.3390/jpm12122018_

Round 1
Reviewer 1 Report
Physical examination has always formed the basis of clinical medicine due to accessibility and low cost. The validation of physical examination results against the standard of imaging tests, surgical or pathological findings is of great importance in clinical medicine. Moreover, the validation of tests for postoperative follow-up is important because it provides the clinician with a simple, effective and inexpensive tool for follow-up without the need for additional imaging tests.
English is cumbersome and requires rewriting by an English-language linguistic editor.
Line 13: Initials should be detailed in the first reference.
Line 60: It is unclear what 'structural damage' is.
The authors state: " The only significant finding was at 3 months post-operative (Chi-Squared: 6.72, p=0.010)". The issue of significance should be given a place in the discussion and conclusion.
The discussion is well written and also details the biomechanical and physiological basis of the findings.
It would be interesting to see if patients who remained positive at the end of the follow-up period showed recurrence of tears or lack of reparative fusion on reimaging.
Author Response
Response to Reviewer 1 Comments
(red=comment to the editor/reviewer; blue=changes in the manuscript)
COMMON
Point 1: English is cumbersome and requires rewriting by an English-language linguistic editor.
Response 1:
Thank you for this feedback. Please note, the last/senior author is a native English speaker and was responsible for the grammar and vocabulary of the manuscript. Nevertheless, we have re-reviewed the manuscript and revised according to your comment.
ABSTRACT
Point 2: Line 13: Initials should be detailed in the first reference.
Response 2:
Thank you for this valuable comment. We revised the title (first mention of the test in the manuscript) as suggested:
@ line 2-4: A Longitudinal Analysis of the Internal Rotation & Shift (IRO/Shift) Test Following Arthroscopic Repair of Superior Rotator Cuff Lesions
MATERIALS and METHODS
Point 3: It is unclear what 'structural damage' is.
Response 3:
As suggested, we have added a short explanation as follows:
@ line 74-85: All subjects were 18 years of age or older (age range: 20.9 – 74.7 years) and presented with a radiologically confirmed structural lesion of the superior rotator cuff, pain, and persistent malfunction of the glenohumeral joint. Prior to introduction to this study, all subjects had completed a conservative rehabilitation program prescribed by their general physician and conducted under the supervision of a physiotherapist which resulted in no reduction of their shoulder symptoms. During or after this conservative treatment, magnetic resonance imaging (MRI) was then determined to be necessary to detect the extent of structural pathology and the reasoning for their lack of improvement. Upon review of the MRI’s, by an experienced radiologist, structural lesions of the supraspinatus were identified. Following the MRI, all patients were then referred to a single shoulder unit for specialized examination and to determine if surgical intervention was indicated.
RESULTS/ DISCUSSION
Point 4: The authors state: " The only significant finding was at 3 months post-operative (Chi-Squared: 6.72, p=0.010)". The issue of significance should be given a place in the discussion and conclusion.
Response 4:
Thank you for pointing out this omission. To increase the reader’s understanding of this important result, we have added several short paragraphs in the statistical section, results section and in the discussion as follows:
@ line 127-131: The observed accuracy (percentage) of both tests was defined as consistent negative and positive results and assessed using a four-field table. In this context, a Chi-squared analysis was used to determine the relation between positive and negative findings for each test in order to illustrate the recovery process.
@ line 175-178: In relation to the healing process, the number of negative matched findings increased sharply from n=1 (pre-operatively) to n=10 (3-months post-operatively) and n=31 (6-months post-operatively). Conversely, the number of concurrent positive findings decreased from n=45 to n=26 and n=3 during this observation period (Table 2).
@ line 206-219: The differences in stress placed on the rotator cuff tendons by the IRO/Shift and the Jobe tests may potentially explain the divergent correlations of the two tests accuracy related to the time of examination throughout the pathological process (pre- and post-operatively) (Table 2). From a statistical point of view, the observed accuracy between both tests steadily decreased from 90% (pre-operatively) to 71% (3-months post-operatively) and 67% (6-months post-operatively). The only statistically significant relationship was found during the second examination (3-month post-operative) (Chi-Squared: 6.72; p=0.010). From a biological and orthopedic perspective, this relationship at the 3-month post-surgery period may have occurred due to the biological demand placed on the tendon while still healing. Conversely, during the 6-month examination the difference in tissue loads between tests may have been more prominent because of the surgically fixed tendon to the footprint and the capability to resist tensile force produced by the IRO/Shift test and the lack of contractile muscle strength generated by an active movement of the arm during the Jobe test.
DISCUSSION
Point 5: The discussion is well written and also details the biomechanical and physiological basis of the findings.
It would be interesting to see if patients who remained positive at the end of the follow-up period showed recurrence of tears or lack of reparative fusion on reimaging.
Response 5:
This is a great point and with your permission, we would like to add this aspect/thought in the discussion.
@ line 257-260: Similarly, future research should investigate if patients who still present with a positive IRO/Shift test at the end of the follow-up period (24-months) demonstrate a recurrence of tears or lack of reparative fusion on reimaging.
Please note, we also improved the introduction, results and conclusion (all changes blue marked) in order to enhance the quality of the manuscript including to increase the word account as suggested by the editor.
Reviewer 2 Report
Very interesting study on clinical shoulder examination and a great addition to the recently published IRO/Shift test. Well discussed findings.
Author Response
Response to Reviewer 2 Comments
(red=comment to the editor/reviewer; blue=changes in the manuscript)
COMMON
Point 1: Very interesting study on clinical shoulder examination and a great addition to the recently published IRO/Shift test. Well discussed findings.
Response 1:
Thank you for this positive feedback and appreciation of our work.